# A Systematic Review and Meta-Analysis of the Inflammatory Biomarkers in Mild Traumatic Brain Injury

**DOI:** 10.3390/biomedicines12020293

**Published:** 2024-01-26

**Authors:** Ioannis Mavroudis, Alin Ciobica, Ioana Miruna Balmus, Vasile Burlui, Laura Romila, Alin Iordache

**Affiliations:** 1Department of Neurology, Leeds Teaching Hospitals, NHS Trust, Leeds LS2 9JT, UK; 2Department of Biology, Faculty of Biology, Alexandru Ioan Cuza University, 700506 Iasi, Romania; 3Department of Exact Sciences and Natural Sciences, Institute of Interdisciplinary Research, “Alexandru Ioan Cuza” University of Iași, 700057 Iasi, Romania; 4Preclinical Department, Apollonia University, Păcurari Street 11, 700511 Iasi, Romania; 5Faculty of Medicine, University of Medicine and Pharmacy “Grigore T. Popa”, 700115 Iasi, Romania

**Keywords:** mild traumatic brain injury, post-concussion syndrome, neuroinflammation, proinflammatory cytokines, diagnosis, outcomes, recovery

## Abstract

Mild traumatic brain injury (mTBI) accounts for most TBI cases, the leading cause of morbidity and mortality worldwide. Despite its high incidence, mTBI pathophysiology remains largely unknown. Recent studies have shown that the inflammatory response is activated early after mTBI and can persist for several weeks or months. However, limited evidence on the utility of inflammatory biomarkers as predictors of clinical outcomes in mTBI has been previously provided. Thus, this systematic review and meta-analysis aims to provide an overview of the current knowledge on the role of inflammation in the pathogenesis of mTBI and the potential of some inflammatory biomolecules as biomarkers of mTBI. In this regard, eight studies comprising 1184 individuals were selected. Thus, it was shown that the increase in IL-6, TNF-α, and IL-1β plasma levels could be implicated in the development of early post-concussion symptoms. On the other hand, the persistence of the increased plasmatic concentrations of IL-10 and IL-8 for as long as six months following the brain injury event could suggest chronic inflammation leading to neuroinflammation and late or persistent symptoms. In this context, our findings showed that inflammatory biomarkers could be relevant in diagnosing or predicting recovery or long-term outcomes of mTBI.

## 1. Introduction

Traumatic brain injury (TBI) is a leading cause of morbidity and mortality worldwide. Mild TBI (mTBI), also known as concussion, accounts for most TBI cases and is characterized by a transient alteration of consciousness or cognitive function. Despite the high incidence of mTBI, its pathophysiology remains largely unknown. In recent years, there has been a growing interest in the role of inflammation in the pathogenesis of mTBI [1].

The prevalence of concussion varies across different populations and settings. In the general population, the estimated prevalence of concussion ranges from 1.6% to 3.8% [2] with higher values in specific high-risk populations, such as athletes and military personnel [3,4]. The prevalence of concussions in high school athletes has been estimated to be as high as 11.2% [5].

The inflammatory response is a complex process that involves the activation of various cell types, including microglia and astrocytes, and the release of a variety of proinflammatory and anti-inflammatory mediators. Studies have shown that the inflammatory response is activated early after mTBI and can persist for several weeks or months. Several inflammatory biomarkers have been proposed as potential markers of mTBI, including interleukin-1 (IL-1), interleukin-6 (IL-6), tumor necrosis factor-alpha (TNF-α), and C-reactive protein (CRP). Recent studies showed that these molecules could be found in increased concentrations in the serum, cerebrospinal fluid (CSF), and saliva of individuals who have undergone mTBI, in a severity- and outcome-dependent manner [6,7,8]. While inflammation is characterized by the activation of immune cells and the release of inflammatory mediators, such as cytokines and chemokines [9], in the context of concussion, it has been proposed as a possible contributor to the pathophysiology of the injury and persistence of symptoms in some individuals [10].

Several studies have investigated the levels of inflammatory biomarkers in individuals with concussions and reported controverted results. While some studies reported increased levels of cytokines (IL-6 and TNF-α) in the serum or CSF [7,8], others did not observe significant differences in concussed individuals, as compared to age- and sex-matched controls [11,12]. In this context, limited evidence on the utility of inflammatory biomarkers as predictors of clinical outcomes in concussion was provided. Several studies suggested that higher levels of inflammatory biomarkers may be associated with worse clinical outcomes, such as prolonged recovery or persistent symptoms [8,13], whereas other studies failed to confirm this association [11,12].

Thus, this study aims to provide an overview of the current knowledge on the role of inflammation in the pathogenesis of mTBI and the potential of some inflammatory biomolecules as biomarkers of mTBI. While summarizing the studies investigating inflammatory biomarkers in mTBI in diagnosis, prognosis, and therapy, we will also discuss the limitations of the current literature and future directions for research in this field.

## 2. Materials and Methods

The aim of this meta-analysis was to examine the evidence on the association between inflammatory biomarkers and mTBI, as well as the potential utility of inflammatory biomarkers as diagnosis and prognosis tools in mTBI.

### 2.1. Search Strategy

A systematic literature search was conducted using the PubMed, Web of Science, Scopus, Embase, and Cochrane Library databases. The search was limited to articles published in English from January 2000 to December 2022. The search terms used were “mild traumatic brain injury” OR “traumatic brain injury” OR “mTBI” OR “TBI” OR “concussion” AND “inflammatory biomarkers” OR “cytokines” OR “chemokines” OR “CRP” OR “IL-1” OR “IL-6” OR “TNF-alpha”. The reference lists of identified studies were also searched for additional relevant studies. This review was performed in accordance with the PRISMA (Preferred Reporting Items for Systematic Reviews and Meta-Analyses) guidelines.

### 2.2. Selection Criteria

Inclusion criteria for this meta-analysis were (1) observational or interventional studies that examined the association between inflammatory biomarkers and mTBI or concussion; (2) studies that measured inflammatory biomarkers in serum or plasma; and (3) studies that were written in English.

The exclusion criteria considered were (1) case reports, case series, or reviews; (2) studies that did not measure inflammatory biomarkers in serum or plasma; and (3) studies that were not written in English or that were not available in full text. Also, animal studies and studies in which the full data were not available were not included in our study.

### 2.3. Data Extraction and Quality Assessment

Two reviewers independently screened the abstracts and full-text articles for eligibility and extracted data from eligible studies. Any discrepancies were resolved by mutual consensus. The following data were extracted from each study: first author, year of publication, study design, sample size, inflammatory biomarkers measured, and main findings.

The quality of the included studies was assessed using the Newcastle–Ottawa scale (NOS) for observational studies and the Cochrane Risk of Bias tool for randomized controlled trials (RCTs).

### 2.4. Data Synthesis and Analysis

Descriptive statistics were used to summarize the characteristics of the included studies. The effect sizes of the associations between inflammatory biomarkers and mTBI or concussion were calculated using standardized mean differences (SMDs) and 95% confidence intervals (CIs).

Heterogeneity among the studies, as indicated by the I2 statistic, prompted us to perform additional analyses for a more comprehensive understanding. We also conducted further meta-regression, subgroup, and sensitivity analyses in an attempt to identify the sources of this heterogeneity and to better interpret our results.

Statistical analysis was conducted using R software (version 3.6.2, R Foundation for Statistical Computing, Vienna, Austria). This included a detailed approach involving random-effects models for assessing heterogeneity, with explanations for the selection of these models. We utilized the I2 statistic for heterogeneity assessment and conducted subgroup analyses for different biomarkers and study designs. The level of statistical significance was taken at *p* < 0.05. Additionally, sensitivity analyses and Egger’s test for publication bias were employed to ensure the robustness of our findings.

Publication bias was assessed using Egger’s test and funnel plots. Sensitivity analyses were conducted by excluding studies with a high risk of bias and using a fixed-effects model.

## 3. Results

### 3.1. Overview of the Studies Included in the Present Meta-Analysis

The initial search provided 92 studies (PubMed—61, Web of Science—43, Scopus—58, Embase—31, and Cochrane Library—0). After excluding duplicates, 61 studies were included in the initial screening procedure. A total of 40 publications were excluded because they did not fulfill the inclusion criteria, and 21 studies were included in the final screening procedure. Eight studies were finally included in the meta-analysis (Figure 1). The study was registered to PROSPERO (ID: CRD42024501843/23.01.2024). There were no significant concerns about publication bias and the quality of the studies (Figure 2A,B).

The studies that were selected [15,16,17,18,19,20,21,22,23] evaluated between 9 and 104 mTBI patients in comparison to orthopedic injury patients or healthy controls. The blood samples were analyzed for protein expression or level changes in apolipoprotein, neuron-specific enolase (NSE), GFAP, IL-1β, IL-2, IL-4, IL-6, IL-7, IL-8, IL-10, IL-12, IL-17A, S100B, tau, TNF-α, CCL2, IFN-γ, fractalkine, NfL, and NSE. The summarized description of the selected studies can be found in Table 1. Parker et al. [15] showed that there were significant differences in IL-6 and tau protein expression across time post-injury, irrespective of clinical outcome. In children with persisting symptoms, IL-8 protein expression significantly differed across time post-injury and the increase in TNF-α expression at one to four days post-injury was seen in children with persisting symptoms, as compared with normal recovery.

Rodney et al. [16] found that IL-6 mean concentration was significantly higher in the repetitive TBI group compared to those with 1–2 TBI or no TBI history. In addition, for participants with a history of TBI, PTSD symptom severity, intrusion, and avoidance were significant predictors of higher IL-6 and IL-10 concentrations, respectively. Thompson et al. [17] showed persistent aging-related differences between control groups in concentrations of four cytokines for up to 6 months. At Day 0, IL-6, IL-8, and fractalkine were higher in the older TBI group compared to the older control and the younger TBI groups, while IL-10 was higher in the older TBI compared to the controls. At Month 1, significantly higher concentrations of IL-8, fractalkine, and TNF-α were seen. At six months post-injury, significantly higher concentrations of IL-6 and IL-8 were seen, while a lower concentration of IL-7 was found in older versus younger TBI groups. Powell et al. [18] found that soldiers with mTBI history had higher NSE concentrations than those without, and lifetime mTBI incidence had significant main effects on NSE and S100B concentrations. Sun et al. [19] showed that the levels of IL-1β, IL-6, and CCL2 were acutely elevated in mTBI patients compared to the controls. Ryan et al. [20] found that all children with TBI had increased IL-6 levels. In mTBI, increased IFN-ã and decreased IL-8, IL-10, IL-17A, and TNF-α were seen. In severe TBI, IFN-ã was decreased compared to the controls. Zhao et al. [21] showed that while both male and female patients had changes in the cerebral brain flow and increased levels of IL-1β and IL-6, female patients demonstrated overexpression of IL-8 and low expression of IL-4. Devoto et al. [22] reported that IL-6 and TNF-α concentrations were higher in the TBI group than in the control group. No significant differences were found in IL-10 or the IL-6/IL-10 ratios between those with low and high PTSD. Vedantam et al. [23] showed that levels of IL-6 and IL-2 within 24 h post-injury were significantly higher for mTBI patients compared to the controls, the latter’s level increase being associated with more severe 1-week post-concussive symptoms. At 6 months post-TBI, increased plasma IL-10 levels were associated with greater depression and more severe PTSD symptoms.

### 3.2. Meta-Analysis

A summarization of the findings described in this section is presented in Table 2.

#### 3.2.1. Interferon-γ

Six studies were included in the meta-analysis on IFN-γ levels in mTBI and PCS, with 546 patients. The statistical analysis showed that the sample was heterogeneous with an I2 of 95.3%, a Q of 107.36, five degrees of freedom, and a *p*-value < 0.0001. A random effects model showed an SMD of 24.3 but with no statistical significance (*p* = 0.21) (Figure 3A). The influence analysis showed that omitting each of the studies and recalculating the SMD was associated with a nonsignificant result, indicating that IFN-γ levels do not differ significantly between the groups of the study (Figure 3B).

#### 3.2.2. TNF-α

Data from nine studies with 687 participants were combined to analyze TNF-α levels. The heterogeneity of the data was high (I2 = 99.6%, Q = 1926.30, d.f. = 7, *p* < 0.000001). We used a random effects model with an SMD of 1.08 and a *p*-value of 0.65. This confirmed that there was no statistically significant difference in the levels of mTBI and control patients (Figure 4).

#### 3.2.3. IL-2

Only one study investigated the difference in IL-2 levels between normal controls and patients with mTBI. Vedantam et al. [23] reported an increase up to 6.93 (s.d. 4.65) in mTBI patients compared to the levels of 4.88 (s.d. 3.33) in normal controls.

#### 3.2.4. IL-6

For IL-6, we analyzed data from nine studies and 795 participants. The I2 was 98.4% (Q = 506.71, d.f. = 8, *p* < 0.0001). The random effects model analysis showed an SMD of 4.47, with a *p*-value of 0.013 (Figure 5A). Influence analysis showed that the result was robust and remained significant after omitting each of the studies and recalculating the SMD (Figure 5B).

#### 3.2.5. IL-1β

Six studies were included in the analysis of IL-1β levels, with 421 participants. The I2 was 92.7% (Q = 68.27, d.f. = 5, *p* < 0.0001). The random effects model showed an SMD of 2.7 (*p* = 0.0072) (Figure 6A). The sensitivity analysis confirmed the robustness of the results, which remained significant after omitting each of the studies (Figure 6B).

#### 3.2.6. IL-4

Data from five studies were included in the analysis of IL-4, with 469 participants. The heterogeneity of the sample was low, with an I2 of 34.1% (Q = 6.07, d.f. = 4, *p* = 0.19). We used a fixed effects model, which showed an SMD of 0.14 (*p* < 0.0001) (Figure 7A). Sensitivity analysis showed that omitting the study of Ryan et al. [20] changed the SMD significantly, and this study severely influenced the final results. This makes the result less robust (Figure 7B).

#### 3.2.7. IL-8

For IL-8, we analyzed the data from five studies with 469 participants. The I2 was 99.9% (Q = 2764,16, d.f. = 4, *p* < 0.000001), and the random effects model we used showed no significant difference between the groups of the study (SMD = 6.96, *p* = 0.45) (Figure 8A).

#### 3.2.8. IL-10

Interleukin 10 levels were analyzed in seven studies with 608 participants. The heterogeneity of the data was high (I2 = 94.2%, Q = 103,51, d.f. = 6, *p* < 0.0001). The random effects model showed no significant difference between the groups of the study (SMD = 0.16, *p* = 0.53) (Figure 8B).

### 3.3. Persistent PCS and Inflammatory Biomarkers

Only two studies investigated the levels of inflammatory biomarkers in persistent PCs. Ventadam et al. [23] measured this in patients with mTBI and controls without injury. They reported that IL-6 was reduced from 54.26 (s.d. 44.27) to 31.15 (s.d 28.6), IL-2 was increased from 6.93 (s.d. 4.65) to 11.33 (s.d. 6.26), IL-10 was increased from 1.67 (s.d. 1.59) to 9.75 (s.d. 0.86), IL-1b was reduced from 9.7 (s.d.) to 5.62 (s.d. 5.4), TNF-α was reduced from 23.56 (s.d. 24.8) to 10.18 (s.d. 9.53), and IFN-γ was reduced from 95.9 (s.d. 90.73) to 33.1 (s.d. 33.2). Powell et al. [18], on the other hand, reported an increase in IL-6 levels from 1.01 (s.d. 1.16) to 1.08 (s.d. 1.14) at one year after a mTBI. Elevated plasma IL-2 levels at 24 h were also associated with more severe PCS 1 week after the injury. Six months after the injury, elevated plasma IL-10 levels were associated with greater depression scores and more severe PTSD symptoms [18].

### 3.4. Publication Bias

Visual inspection of funnel plots and Egger’s tests showed no publication bias for any of the biomarkers that exhibited significant differences between mTBI and control individuals (Figure 9A—IL-1β, B—IL-4, C—IL-6).

## 4. Discussion

In the present meta-analysis, we investigated the role of inflammatory biomarkers in diagnosing mTBI and the prognosis of PCS: IL-1β, IL-2, IL-4, IL-6, IL-8, IL-10, IFN-γ, and TNF-α. Cytokines are involved in inflammatory mechanisms and play an important role in the pathogenesis of many diseases. Thus, IL-2 is a cytokine that plays a crucial role in the immune system by promoting the proliferation and activation of T cells and other immune cells being produced by various cells, including T cells, B cells, and natural killer cells, in response to antigen stimulation. IL-2 acts on cells that express the high-affinity IL-2 receptor (IL-2R) to promote cell growth, differentiation, and effector function. IL-2 has been implicated in a variety of physiological and pathological processes, including immune responses, autoimmunity, transplantation, cancer, and infectious diseases [24]. Dysregulated IL-2 function can lead to various immunological disorders, such as immunodeficiency, autoimmune diseases, or chronic inflammation.

IL-1β is a proinflammatory cytokine primarily produced by activated macrophages and monocytes in response to infection, injury, or stress. IL-1β is involved in various physiological and pathological processes, including inflammation, fever, tissue repair, and immunity, acting on various target cells, including endothelial cells, fibroblasts, and immune cells, by binding to the IL-1 receptor (IL-1R) and activating intracellular signaling pathways. IL-1β promotes the production of other proinflammatory cytokines, such as TNF-α and IL-6, and induces the expression of adhesion molecules and chemokines, which attract immune cells to the site of injury or infection [25]. Changes in IL-1β functions have been implicated in various diseases, including rheumatoid arthritis, type 2 diabetes, and Alzheimer’s disease.

IL-10 is an anti-inflammatory cytokine produced by T cells, B cells, macrophages, and dendritic cells. IL-10 plays a key role in regulating the immune response by inhibiting the production of proinflammatory cytokines, such as TNF-α and IL-1β, and promoting the differentiation and activation of regulatory T cells. IL-10 is involved in a wide range of physiological and pathological processes, including autoimmune diseases, allergies, infections, and cancer. Differences in IL-10 expression or function have been implicated in various immunological disorders, such as immunodeficiency, chronic inflammation, or cancer progression [26].

IL-4 is a cytokine primarily produced by T helper 2 cells (Th2), mast cells, and basophils. It regulates the immune response by promoting the differentiation and activation of Th2 cells, inducing the production of IgE antibodies, and suppressing the activity of Th1 cells and macrophages [27]. IL-4 is involved in several physiological and pathological processes, including allergies, asthma, autoimmunity, and infectious diseases. Dysregulated IL-4 expression or function has been implicated in various immunological disorders, such as allergic diseases, immunodeficiency, and autoimmune diseases.

Interleukins 6 and 8 are also proinflammatory cytokines that are primarily produced by immune cells, including macrophages, monocytes, and T cells, in response to infection, inflammation, or injury [28,29]. IL-6 acts on various target cells, including hepatocytes, lymphocytes, and endothelial cells, by binding to the IL-6 receptor (IL-6R) and activating intracellular signaling pathways. IL-6 promotes the production of acute-phase proteins, such as CRP, and the differentiation and activation of immune cells, such as Th17 cells and B cells. Various inflammatory or immune-mediated diseases, such as rheumatoid arthritis, multiple sclerosis, and cancer, were previously characterized by impaired IL-6 activity [28].

TNF-α and IFN-γ are proinflammatory cytokines primarily produced by activated immune cells, including macrophages, T cells, and natural killer cells, in response to infection, inflammation, or injury. TNF-α acts on various target cells, including endothelial cells, macrophages, and immune cells, by binding to the TNF receptor (TNFR) and activating intracellular signaling pathways. TNF-α promotes the production of other proinflammatory cytokines, such as IL-1β and IL-6, and induces the expression of adhesion molecules and chemokines, which attract immune cells to the site of injury or infection. TNF-α activity impairments have been reported in various inflammatory or immune-mediated diseases, such as rheumatoid arthritis, inflammatory bowel disease, and cancer [30,31].

IFN-γ is a cytokine involved in regulating the immune response by promoting the differentiation and activation of Th1 cells, natural killer cells, and macrophages. It was shown to inhibit the proliferation and activation of Th2 cells and regulatory T cells. Dysregulated IFN-γ functions have been implicated in various immunological disorders, such as autoimmune diseases, chronic infections, and cancer [32].

In the present meta-analysis, we underline other characteristics of the cytokines and their involvement in the pathophysiology of TBI. Thus, recent research data have suggested a potential role of IL-2 in the pathophysiology of TBI and its associated symptoms. Data from one available study present a significant difference in IL-2 serum levels between controls and mTBI patients and also, at six months post-injury, a significant increase in IL-2 serum levels. IL-2 is involved in acute inflammatory response, contributing to the development of early symptoms and long-term recovery. IL-2 could be an important inflammatory biomarker of TBI diagnosis and PCS prognosis.

IL-1β has also been implicated in the pathophysiology of TBI and its associated symptoms [25]. The studies from this meta-analysis suggest that the significantly increased IL-1β levels reported in mTBI patients are associated with the acute phase of inflammatory response and could indicate its role in the pathophysiology of mTBI. Thus, IL-1β is a good candidate as a diagnosis biomarker. Furthermore, it was shown that IL-1 levels decreased six months after the injury and this fact suggested a good potential in prognosis and outcome prediction of the brain injury.

The potential role of IL-10 in the pathophysiology of traumatic brain injury (TBI) and its associated symptoms has been investigated by several studies. However, the present meta-analysis failed to show a significant difference in the expression of IL-10 in patients with an mTBI compared to control individuals. The role of this cytokine as a diagnostic or prognostic biomarker in mTBI seems to be insignificant.

The studies from our meta-analysis show that, although the levels of IL-4 were reduced in mTBI patients and the result was statistically significant, sensitivity analysis failed to confirm the robustness of the result. Thus, further studies are needed to confirm the role of this cytokine as a diagnostic or prognostic biomarker in mTBI [27]. The levels of IL-6 were significantly different between mTBI patients and control individuals, a result that was also confirmed by the sensitivity analysis. This result indicates the potential role of IL-6 as a diagnostic and prognostic biomarker in mTBI.

The levels of IL-8, however, were not significantly different. On the other hand, no clear information about the possible use of IL-8 was observed when we meta-analyzed the previous reports. Despite this, it is already known that the dysregulation of IL-8 expression or function was seen in chronic obstructive pulmonary disease, cystic fibrosis, and sepsis [33]. The recruitment and activation of neutrophils and other immune cells to the site of infection or injury, as well as endothelial cell and leukocyte activation, concur in the intracellular signaling pathways in which IL-8 participates. Kossmann et al. [34] discussed the possible implication of IL-8 in TBI when they observed that it is released into the CSF following a traumatic event resulting in brain injury. In this way, the nonspecific participation of IL-8 in the TBI pathophysiology could be associated with blood–brain barrier dysfunction and nerve recovery [34]. Notwithstanding, Whalen et al. [35] previously described IL-8 as a potential target in anti-inflammatory therapy and suggested that it participates in the main process undergoing the acute inflammatory component of TBI.

The blood TNF- α has no value as a diagnostic biomarker for mTBI, as we can observe in the present study from the meta-analysis, with no significant difference between mTBI patients and controls.

There was statistical significance in the serum levels of IFN-γ between mTBI and individuals without a history of mTBI. However, the data regarding IFN-γ are rather scarce; thus, further analysis could shed more light on this cytokine’s potential as a diagnosis or prognosis biomarker.

The data on the difference of the serum inflammatory biomarkers at 6 and 12 months post-injury were limited, and no clear conclusions can be made.

A distinct systemic inflammatory response following mTBI, quantifiable within 6 h to 12 months post-injury, interleukin-6 is identified as the most promising biomarker for clinical diagnosis of brain injury, with interleukin-10 as a potential candidate for triaging CT scans [36]. This aligns with our findings on the relevance of IL-6 and IL-10 in mTBI. Vedantam et al., focusing on the profile of plasma inflammatory cytokines after mTBI, revealed significant associations between these cytokines and neuropsychological outcomes. Elevated plasma IL-2 and IL-6 within 24 h post-injury were significantly higher in mTBI patients compared to the controls. Elevated plasma IL-2 at 24 h was associated with more severe post-concussive symptoms, while elevated IL-10 at 6 months correlated with greater depression and PTSD symptoms. This underscores the potential role of inflammation in the pathophysiology of post-traumatic symptoms [23]. Another systematic review and meta-analysis found that within 24 h of injury, patients with mTBI have significantly higher levels of IL-6, IL-1RA, and IFN-γ in their blood compared to healthy controls. One week following the injury, higher levels of MCP-1/CCL2 were noted. The study also confirmed elevated blood levels of IL-6, MCP-1/CCL2, and IL-1β in the mTBI population compared to healthy controls, particularly in the acute stages [37]. This supports our findings regarding the acute elevation of specific cytokines post-injury.

In light of these findings, our study contributes valuable insights into the role of inflammatory biomarkers in mTBI. It not only aligns with the emerging evidence on the systemic inflammatory response and its association with clinical outcomes but also highlights the complex and multifaceted nature of biomarker research in this field. The continued exploration of these biomarkers is crucial for enhancing our understanding of mTBI pathophysiology and improving diagnostic and prognostic capabilities.

Thus, we can recognize the limitations of our meta-analysis, especially regarding the number of studies finally included and the results, which sometimes are divergent, as well as the inevitable risks regarding homogeneity and the risk of bias. In addition, as the dynamics of the changes in inflammatory biomarkers are important in mTBI diagnosis and recovery, we found that the studies included in this analysis have variable moments of reporting post-TBI inflammatory markers levels. However, these changes could also be the result of the characteristics of mTBIs, such as different brain regions that are affected or different mTBI types (direct versus indirect trauma, blasts, punches, accidents), and should be further addressed to evaluate if they significantly influence the inflammatory biomarker dynamics. But, overall, these findings suggest that inflammatory biomarkers may serve as potential diagnostic and therapeutic targets for mTBI. Further research is needed to better understand the role of these biomarkers in the pathophysiology of mTBI and to explore their potential as diagnostic or therapeutic tools. Additionally, identifying specific cytokine profiles associated with different mTBI symptom clusters may improve diagnosis and treatment efficacy.

## 5. Conclusions

Despite the increased incidence of mTBI, there is scarce evidence about its pathophysiology. The implication of inflammation in mTBI was previously shown. However, the inflammatory biomarkers used in diagnosis, prognosis, and therapy need further documentation. This study provides reasonable evidence based on the meta-analysis of several studies presenting the variations of inflammatory cytokines in mTBI. Thus, it was shown that the increase in IL-6, TNF-α, and IL-1β plasma levels could be implicated in the development of early post-concussion symptoms. On the other hand, the persistence of the increased plasmatic concentrations of IL-10 and IL-8 for as long as six months following the brain injury event could suggest chronic inflammation leading to neuroinflammation and late or persistent symptoms. In this context, our findings showed that inflammatory biomarkers could be relevant in diagnosing or predicting recovery or long-term outcomes of mTBI.

## Figures and Tables

**Figure 1 biomedicines-12-00293-f001:**
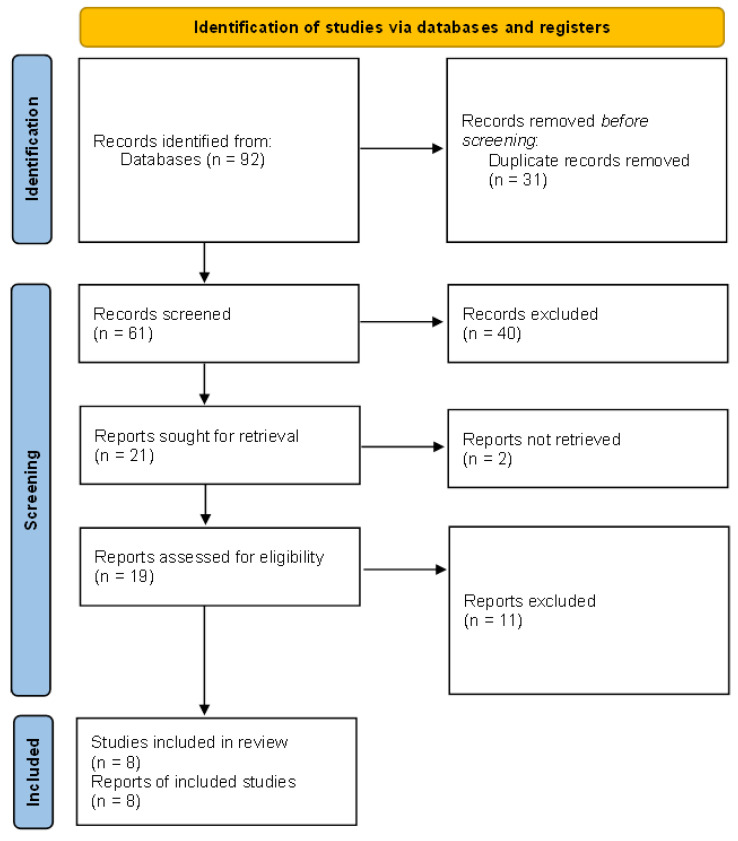
PRISMA 2020 flow diagram for new systematic reviews, which included searches of databases and registers only; according to the guidelines provided by PRISMA [14].

**Figure 2 biomedicines-12-00293-f002:**
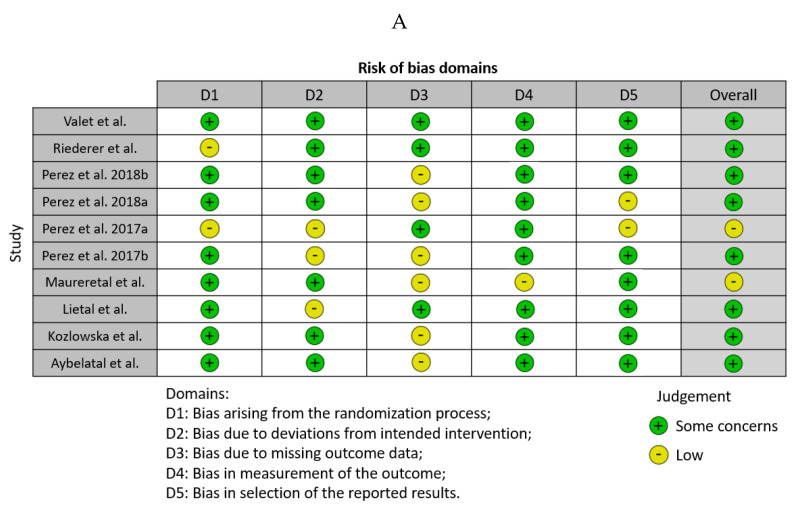
The analysis of publication bias [15,16,17,18,19,20,21,22,23] (**A**) and the quality of the studies (**B**).

**Figure 3 biomedicines-12-00293-f003:**
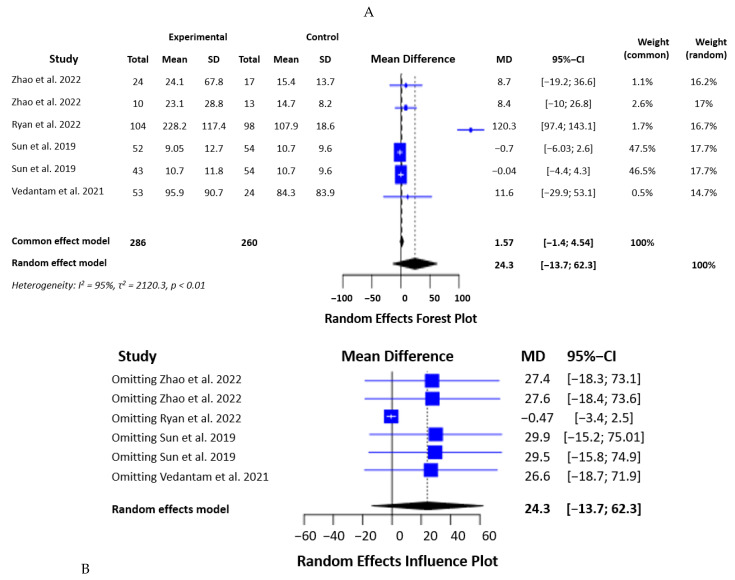
Statistical analysis of the studies included in the meta-analysis on IFN-γ levels in mTBI and PCS (forest plot—(**A**), effects influence plot—(**B**)) [19,20,21,22,23].

**Figure 4 biomedicines-12-00293-f004:**
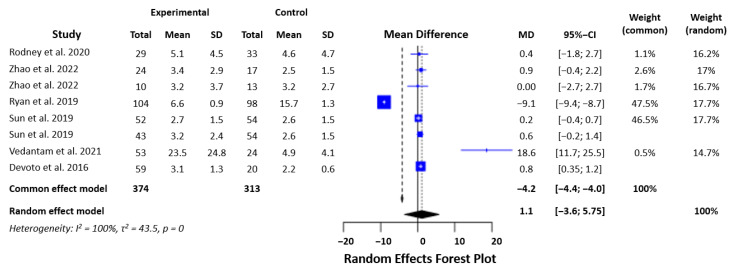
Statistical analysis of the studies included in the meta-analysis on TNF-α levels in mTBI and PCS (forest plot) [16,19,20,21,22,23].

**Figure 5 biomedicines-12-00293-f005:**
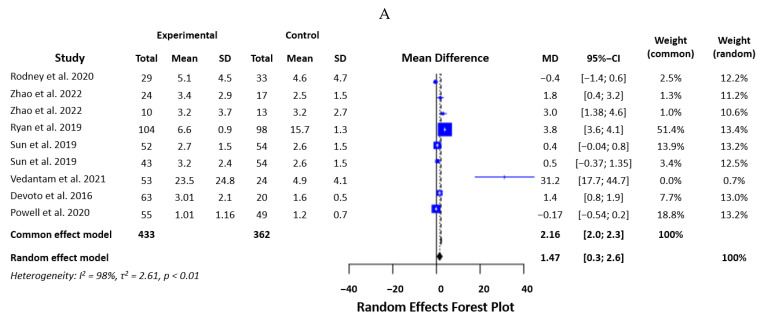
Statistical analysis of the studies included in the meta-analysis on IL-6 levels in mTBI and PCS (forest plot—(**A**), effect influence plot—(**B**)) [16,18,19,20,21,22,23].

**Figure 6 biomedicines-12-00293-f006:**
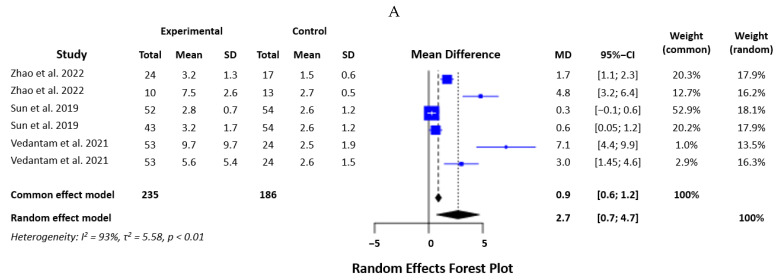
Statistical analysis of the studies included in the meta-analysis on IL-1β levels in mTBI and PCS (forest plot—(**A**), effect influence plot—(**B**)) [19,21,23].

**Figure 7 biomedicines-12-00293-f007:**
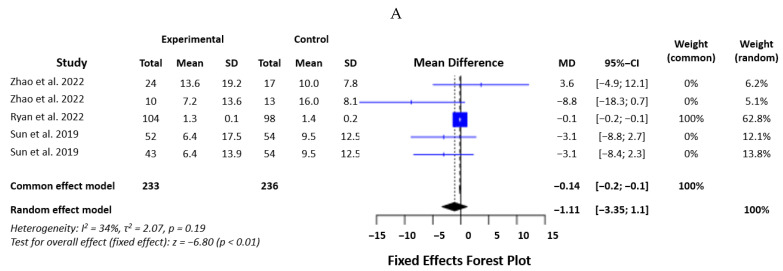
Statistical analysis of the studies included in the meta-analysis on IL-4 levels in mTBI and PCS (forest plot—(**A**), effect influence plot—(**B**)) [19,20,21].

**Figure 8 biomedicines-12-00293-f008:**
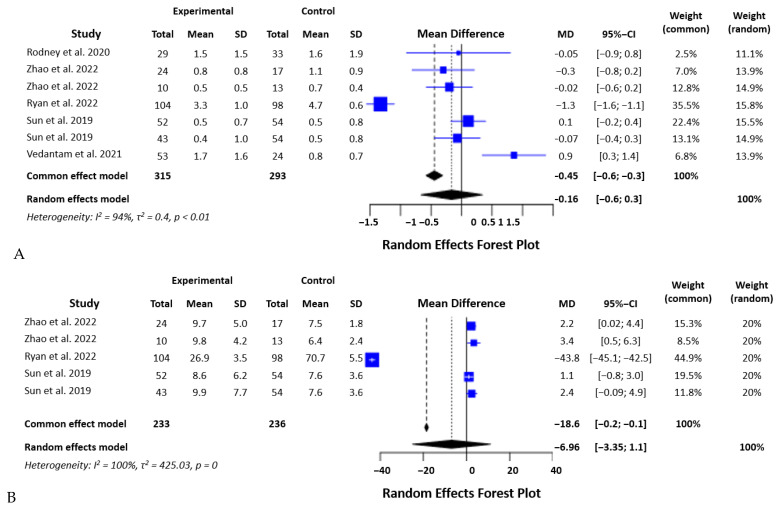
Statistical analysis of the studies included in the meta-analysis (Forest plots) on IL-8 (**A**) and IL-10 (**B**) levels in mTBI and PCS [16,19,20,21,22,23].

**Figure 9 biomedicines-12-00293-f009:**
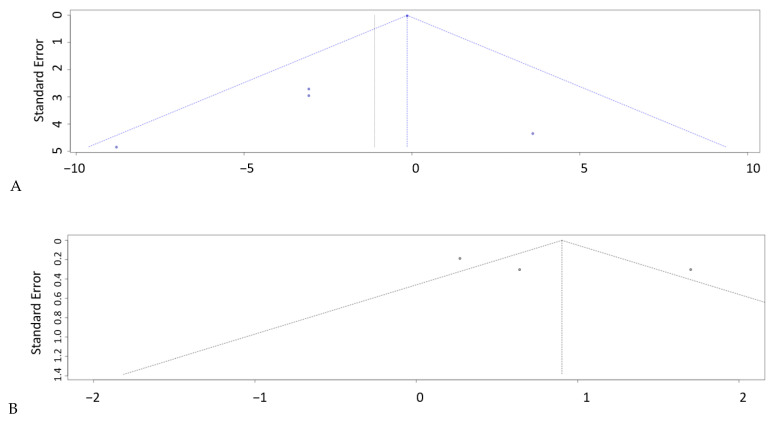
Egger test for (**A**)—IL-1β, (**B**)—IL-4, and (**C**)—IL-6.

**Table 1 biomedicines-12-00293-t001:** Summarized description of the studies that were selected for the meta-analysis.

Study Description	Evaluated Biomarkers	Main Findings	Reference
pediatric mTBI patients: normal recovery (*n* = 9), persistent symptoms (*n* = 9).	TNF-α, IL-1β, IL-6, IL-8, IL-10, S100B, Tau, NSE, GFAP.	significant differences in IL-6 and tau expression following mTBI; significant differences in IL-8 expression in children with persisting symptoms; increased TNF-α expression in children with persisting symptoms, as compared with normal recovery.	[15]
106 participants withhistory of TBI: repetitive TBI (*n* = 44), 1–2 TBIs (*n* = 33), no TBIs (*n* = 29).	TNF-α, IL-6, IL-10.	IL-6 significantly higher in the repetitive TBI group, as compared to the other groups; higher IL-6 and IL-10 concentrations correlated with PTSD symptoms.	[16]
313 participants: young mTBI patients (*n* = 96); older mTBI patients (*n* = 75); age/sex-matched noninjured controls (*n_y_* = 80, *n_o_* = 62).	IL-6, IL-7, IL-8, IL-10, TNF-α, Fractalkine.	increased IL-6, IL-8, and fractalkine in older mTBI patients, as compared to older controls and younger mTBI patients (acute); increased IL-10 in the older mTBI patients, as compared to their controls (acute); increased TNF-α in mTBI patients (1 month post-injury); increased and persistent IL-6 and IL-8 in mTBI patients (6 months post-injury); decreased IL-7 levels in older mTBI patients, as compared to the younger ones.	[17]
104 active-duty combat soldiers grouped by mTBI history, lifetime mTBI incidence.	NfL, NSE, S100B, IL-6.	increased NSE concentrations in soldiers with mTBI history, as compared to those without mTBI history; significant main effects on NSE and S100B concentrations of lifetime mTBI incidence; significant main effect on NfL concentration of mTBI recurrence.	[18]
106 adults mTBI patients (*n* = 52, *n* = 43), matched healthy control (*n* = 54).	IL-1β, IL-4, IL-6, IL-8, IL-10, IL-12, CCL2, IFN-γ, TNF-α.	IL-1β, IL-6, and CCL2 significantly increased in mTBI patients, as compared to controls; CCL2 increase in direct correlation with PCS severity; IL-1β increase in inverse correlation with working memory performance.	[19]
208 children severe TBI (initial GCS ≤ 8) (*n* = 6), mTBI (GCS 14/15) (*n* = 104), healthy controls (*n* = 98).	IL-6 IL-8, IL-10, IL-17A, TNF-α, IFN-γ.	increased IL-6 in mTBI patients, as compared to controls; decreased IL-8, IL-10, IL-17A, and TNF-α in mTBI, as compared to controls; significantly increased IFN-γ in mTBI patients, as compared to controls; significantly decreased IFN-γ in severe TBI patients, as compared to controls.	[20]
41 patients with acute stage mTBI: 17 females, 24 males; 23 sex-, age-, and education-matched healthy participants: 13 females, 10 males.	IL-1β, IL-4, IL-6, IL-8.	increased levels of IL-1β and IL-6 after mTBI; overexpression of IL-8 and low expression of IL-4 in females that had undergone mTBI.	[21]
83 individuals: 63 active-duty personnel with and without a history of TBI, 20 age/sex-matched controls.	TNF-α, IL-6, IL-10.	increased IL-6 and TNF-α in the TBI patients, as compared to control group; positive correlation between IL-6 and TNF-α and PTSD severity.	[22]
157 individuals: 104 patients with mTBI, 53 orthopedic injury controls.	IL-2, IL-6, IL-10.	increased plasma IL-2 and IL-6 levels in mTBI patients (24 h after the injury), as compared to controls; significant correlation between plasma IL-2 levels (24 h) and 1-week PCS severity; significant correlation between plasma IL-10 (6 months) and PTSD symptoms severity.	[23]

**Table 2 biomedicines-12-00293-t002:** Summarized description of the results of the concentration of cytokines presented in the studies from the meta-analysis.

Cytokine	No. of Studies from Meta-Analysis No. of Participants in the Studies	Results of the Studies (mTBI vs. Control Groups) Cytokine Concentration
IFN-γ	6 studies, 546 participants	No significant differences between groups
TNF-α	9 studies, 687 participants	No significant differences between groups
IL2	1 study, 77 participants	Increased level in mTBI group
IL6	9 studies, 795 participants	Increased level in mTBI group
IL1β	6 studies, 421 participants	Increased level in mTBI group
IL4	5 studies, 469 participants	Increased level in mTBI group
IL8	5 studies, 469 participants	Increased level in mTBI group
IL10	7 studies, 608 participants	Increased level in mTBI group

## Data Availability

The data presented in this study are available upon reasonable request from the corresponding author.

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
