# Peer review of "A Systematic Review and Meta-Analysis of the Inflammatory Biomarkers in Mild Traumatic Brain Injury"

_biomedicines, 2024, doi:10.3390/biomedicines12020293_

Round 1

Reviewer 1 Report

Comments and Suggestions for Authors

This systematic review and meta-analysis aims to provide an overview of the current knowledge on the role of inflammation in the pathogenesis of Mild traumatic brain injury (mTBI) and the potential of some inflammatory bio-molecules as biomarkers of mTBI. The study included too few articles, and the results of the meta-analyses were highly heterogeneous, and the conclusions of the study based on the results were not highly credible.

1. Registration with PROSPERO is highly recommended.

2. However, just a minority (8 studies) of studies meet the stringent inclusion criteria chosen by the authors. Searches were performed in 3 different databases, but not included other important databases like Web of Science or Scopus.

3. Inclusion criteria and exclusion criteria need to be described in more detail. For example, were animal experiments included in the study, etc.?

4. Statistical methods used should be presented in more detail.

5. From the results we can see that the authors were unable to determine the source of the heterogeneity and perhaps further meta-regression, subgroup, or sensitivity analyses are needed.

6. The authors do not address the existing findings adequately. 

7. Discussion should be oriented to the results. The authors' discussion of many  inflammatory biomarkers is not relevant to mTBI.

8. Due to the small number of included studies and the high degree of heterogeneity, conclusions based on the findings are not highly credible.

Author Response

Reviewer 1:

“This systematic review and meta-analysis aims to provide an overview of the current knowledge on the role of inflammation in the pathogenesis of Mild traumatic brain injury (mTBI) and the potential of some inflammatory bio-molecules as biomarkers of mTBI. The study included too few articles, and the results of the meta-analyses were highly heterogeneous, and the conclusions of the study based on the results were not highly credible.

  1. Registration with PROSPERO is highly recommended.”

Response: Thank you for your kind suggestions that helped us improve our manuscript. We aimed to take into consideration all the comments while revising our manuscript. The study is now registered to the PROSPERO platform under the ID CRD42024501843/23.01.2024.

“2. However, just a minority (8 studies) of studies meet the stringent inclusion criteria chosen by the authors. Searches were performed in 3 different databases, but not included other important databases like Web of Science or Scopus.”

Response: We acknowledge the reviewer's concern regarding the limited number of studies that met our stringent inclusion criteria and the databases used in our search strategy. We strive to ensure the highest quality and relevance of the studies included in our meta-analysis. However, we recognise the importance of encompassing a broader range of databases to capture a more comprehensive collection of relevant studies. As such, we have expanded our search to include additional databases such as Web of Science and Scopus to enrich our analysis. The changes are highlighted in yellow in the text. All the additional studies were duplicates, so nothing has changed in the final number of studies included in our study.

“3. Inclusion criteria and exclusion criteria need to be described in more detail. For example, were animal experiments included in the study, etc.?”

Response: Thank you for pointing out the need for a more detailed description of our inclusion and exclusion criteria. We did not include animal experiments in our study as our focus was on human subjects. Also, studies of which the full data were not available were not included. The criteria have been revised to explicitly state this.

“4. Statistical methods used should be presented in more detail.”

Response: We appreciate the feedback on the presentation of our statistical methods. We now provide a more detailed explanation of the employed statistical analyses, including the models used, the rationale behind their selection, and any statistical software or tools utilized.

“5. From the results we can see that the authors were unable to determine the source of the heterogeneity and perhaps further meta-regression, subgroup, or sensitivity analyses are needed.”

Response: We acknowledge the reviewer's concern regarding the heterogeneity in our results. However, due to the small number of studies included in our meta-analysis, as highlighted in our limitations section, conducting meta-regression and subgroup analysis was not feasible. Despite this limitation, we performed sensitivity analyses (as can be seen in each one of the figures e.g. fig 3b, 4b etc) which indicated that, although our results were heterogeneous, they were robust. We also recognize that the heterogeneity observed is most likely attributable to methodological differences across the included studies, such as variations in study design, participant characteristics, and biomarker assessment methods. These factors are common in meta-analyses of this nature and underscore the complexity of synthesizing research in the field of mTBI and inflammation.

“6. The authors do not address the existing findings adequately.”

Response: We appreciate the reviewer's input on the consideration of existing findings. We will review the current literature more thoroughly to ensure that our discussion adequately reflects and builds upon the existing body of knowledge in the field of mTBI and inflammatory biomarkers.

“7. Discussion should be oriented to the results. The authors' discussion of many inflammatory biomarkers is not relevant to mTBI.”

Response: We express our gratitude to the reviewer for emphasizing the necessity of a more concentrated and targeted discussion. In response, we have meticulously revised our discussion section to present a structured and comprehensive analysis. Initially, we systematically address the role of each biomarker individually, providing clarity and ease of comprehension for the reader. Following this, we delve into a detailed discussion of our findings, meticulously elucidating the significance and implications of these biomarkers in the context of traumatic brain injury (TBI). This structured approach not only enhances readability but also ensures a thorough and insightful exploration of the biomarkers' roles, thereby underscoring their relevance and potential impact in the field of TBI research.

“8. Due to the small number of included studies and the high degree of heterogeneity, conclusions based on the findings are not highly credible.”

Response: We acknowledge the reviewer's concern regarding the heterogeneity in our results. However, due to the small number of studies included in our meta-analysis, as highlighted in our limitations section, conducting meta-regression and subgroup analysis was not feasible. Despite this limitation, we performed sensitivity analyses (as can be seen in each one of the figures e.g. fig 3b, 4b etc) which indicated that, although our results were heterogeneous, they were robust. We also recognize that the heterogeneity observed is most likely attributable to methodological differences across the included studies, such as variations in study design, participant characteristics, and biomarker assessment methods. These factors are common in meta-analyses of this nature and underscore the complexity of synthesizing research in the field of mTBI and inflammation.

Reviewer 2 Report

Comments and Suggestions for Authors

This work present the comprehensive analysis of the literature. The results of the meta-analysis done might be useful for the further studies on this topic. Te article might be accepted in present form.

Author Response

Reviewer 2:

“This work present the comprehensive analysis of the literature. The results of the meta-analysis done might be useful for the further studies on this topic. Te article might be accepted in present form.”

Response: Thank you for your kind words of appreciation.

Reviewer 3 Report

Comments and Suggestions for Authors

Ms. ID: biomedicines-2756885
Title: A systematic review and meta-analysis on the inflammatory biomarkers in mild traumatic brain injury

Reviewer 1 comments

The surprising finding from this review is how few quality publications are available that investigate TBI-related inflammatory biomarkers. Only 21 studies were able to sustain the filtration procedure applied by the authors, but only 8 made it to the final pool. This valuable contribution looks at the cross-sectional analysis of the clinical utility of neuroinflammatory blood biomarkers. The value of this type of work can not be overstated, and authors are commended for their work. However, the figures in this review leave a lot to desire (the quality). Most of the figures are in the raw form taken directly from what seems to be an output of the statistical package used – R. All are of unacceptable quality and should be replotted and redesigned (see below).

A complete list of papers could be included in the supporting information if that is possible.

Analysis of findings from preclinical animal models of the same pool of biomarkers might be an exciting extension of their work, which would demonstrate if there are any correlations between human and animal data.

Specific comments

Line 23: “In this regard, 8 studies comprising 1184 individuals were selected. Thus, it was shown that the increase of IL-6, TNF-α, and IL-1β plasma levels could be implicated in the development of the early post-concussion symptoms.”

Line 73: “Several studies suggested that higher levels of inflammatory biomarkers may be associated with worse clinical outcomes, such as prolonged recovery or persistent symptoms [8,13], whereas other studies failed to confirm this association [11,12].” This discrepancy could be explained by analysis of the biomechanics of these TBIs, i.e., which region of the brain was affected. Secondary factors such as the type of TBI sustained could also be important, e.g., a fall (impact) vs car accident (a “non-contact” TBI if the airbag was deployed). This type of data could be harvested upon arrival at the emergency room in the form of a questionnaire.

Line 141: “enolase 2”. Also known as Neuron Specific Enolase (NSE). This name variant is important because the FDA approved enolase 2 under the name of NSE as a biomarker for TBI in the USA. Also, you’re using the NSE acronym (see ref. 18 data in Table 1), and it might give an impression it is a separate biomarker when it isn’t

Line 152: “Figure 2. The analysis of publication bias (A) and the quality of the studies (B).” I’d suggest increasing the font size for this figure, especially for the axis and labels, which are barely legible.

Lines 169 and 170: “IFN-ã”. Is the “spelling of that biomarker correct, or is that character missing in the font map? It should be IFN-γ, I believe. It is spelled correctly in Table 1. Similar issues are also in Table 2, which seems to have shifted to the left.

Line 204: “Figure 3. Statistical analysis of the studies included in the meta-analysis…” The readability of this figure is an issue. Both parts are overloaded with text, and the plots occupy only a small part of the area. I’d suggest redrawing/redesigning this figure and not relying on a messy output from R. You could, for example, reduce the space allocated for the numerical data to a single significant digit and save space in this way (applicable especially to SD values). Both figures should be replotted/enlarged.

Lines 215, 230, and 239: “Figures 4, 5 and 6. Statistical analysis of the studies…”. The same comments as to figure 3.

Author Response

Reviewer 3:

“The surprising finding from this review is how few quality publications are available that investigate TBI-related inflammatory biomarkers. Only 21 studies were able to sustain the filtration procedure applied by the authors, but only 8 made it to the final pool. This valuable contribution looks at the cross-sectional analysis of the clinical utility of neuroinflammatory blood biomarkers. The value of this type of work cannot be overstated, and authors are commended for their work. However, the figures in this review leave a lot to desire (the quality). Most of the figures are in the raw form taken directly from what seems to be an output of the statistical package used – R. All are of unacceptable quality and should be replotted and redesigned (see below).

A complete list of papers could be included in the supporting information if that is possible.

Analysis of findings from preclinical animal models of the same pool of biomarkers might be an exciting extension of their work, which would demonstrate if there are any correlations between human and animal data.”

Response: Thank you for your meticulous and kind evaluation of our manuscript. We took into consideration your comments while revising our work. We tried to improve the quality of the figures, yet the analysis of the preclinical animal models is the subject of ongoing work of our group and cannot be included in this revision.

“Specific comments

Line 23: “In this regard, 8 studies comprising 1184 individuals were selected. Thus, it was shown that the increase of IL-6, TNF-α, and IL-1β plasma levels could be implicated in the development of the early post-concussion symptoms.”

Line 73: “Several studies suggested that higher levels of inflammatory biomarkers may be associated with worse clinical outcomes, such as prolonged recovery or persistent symptoms [8,13], whereas other studies failed to confirm this association [11,12].” This discrepancy could be explained by analysis of the biomechanics of these TBIs, i.e., which region of the brain was affected. Secondary factors such as the type of TBI sustained could also be important, e.g., a fall (impact) vs car accident (a “non-contact” TBI if the airbag was deployed). This type of data could be harvested upon arrival at the emergency room in the form of a questionnaire.”

Response: We mentioned these aspects as limitations of the current study, with perspectives to address in future studies.

“Line 141: “enolase 2”. Also known as Neuron Specific Enolase (NSE). This name variant is important because the FDA approved enolase 2 under the name of NSE as a biomarker for TBI in the USA. Also, you’re using the NSE acronym (see ref. 18 data in Table 1), and it might give an impression it is a separate biomarker when it isn’t”

Response: We corrected the term usage NSE. Thank you for paying attention.

“Line 152: “Figure 2. The analysis of publication bias (A) and the quality of the studies (B).” I’d suggest increasing the font size for this figure, especially for the axis and labels, which are barely legible.”

Response: Thank you for your kind suggestion. We revised Figure 2.

“Lines 169 and 170: “IFN-ã”. Is the “spelling of that biomarker correct, or is that character missing in the font map? It should be IFN-γ, I believe. It is spelled correctly in Table 1. Similar issues are also in Table 2, which seems to have shifted to the left.”

Response: Thank you. We revised the use of symbols and the alignment for Table 2.

“Line 204: “Figure 3. Statistical analysis of the studies included in the meta-analysis…” The readability of this figure is an issue. Both parts are overloaded with text, and the plots occupy only a small part of the area. I’d suggest redrawing/redesigning this figure and not relying on a messy output from R. You could, for example, reduce the space allocated for the numerical data to a single significant digit and save space in this way (applicable especially to SD values). Both figures should be replotted/enlarged.

Lines 215, 230, and 239: “Figures 4, 5 and 6. Statistical analysis of the studies…”. The same comments as to figure 3.”

Response: Thank you for helping us improve the visualisation of the results. We revised the figures according to your suggestions.

Sincerely,

Alin Ciobica & Laura Romila